# Bicyclo [6.3.0] Undecane Sesquiterpenoids: Structures, Biological Activities, and Syntheses

**DOI:** 10.3390/molecules24213912

**Published:** 2019-10-30

**Authors:** Guo-Fei Qin, Hong-Bao Liang, Wen-Xiu Liu, Feng Zhu, Ping-Lin Li, Guo-Qiang Li, Jing-Chun Yao

**Affiliations:** 1State Key Laboratory of Generic Manufacture Technology of Chinese Traditional Medicine, Lunan Pharmaceutical Group Co. Ltd., Linyi 273400, China; qinguofei@126.com (G.-F.Q.); lianghongbao1985@163.com (H.-B.L.); nwzf@nwafu.edu.cn (F.Z.); 2Key Laboratory of Marine Drugs, Chinese Ministry of Education, School of Medicine and Pharmacy, Ocean University of China, Yushan Road 5, Qingdao 266003, China; lipinglin@ouc.edu.cn; 3Laboratory of Marine Drugs and Biological Products, National Laboratory for Marine Science and Technology, Qingdao 266235, China; 4Jiangsu Hengrui Pharmaceutical Co. Ltd., Lianyungang 222002, China; liuwenxiu08@126.com

**Keywords:** sesquiterpenoids, bicyclo [6.3.0], five-eight-membered ring, cyclooctane, structures, biological activities, syntheses

## Abstract

Sesquiterpenoids constitute a marvelously varied group of natural products that feature a vast array of molecular architectures. Among them, the unusual bicyclo [6.3.0] undecane sesquiterpenoids are one of the most representative. To date, only approximately 42 naturally occurring compounds with this unique scaffold, which can be classified into seven different groups, have been reported. As the first-found member of each type, dactylol, asteriscanolide, dumortenol, toxicodenane C, and capillosanane S are characteristic of the four methyl groups on the five-eight-membered ring system. Only 11-hydroxyjasionone and sinuketal decorate the core with an isopropyl group. These natural products exhibit a wide range of bioactivities, including antifouling, anti-inflammatory, immune suppression, cytotoxic, antimutagenic, antiplasmodial, and antiviral activities. It was noted that some total syntheses of precapnellane-sesquiterpenoids (dactylol, poitediol, precapnelladiene), asteriscanolide, and 11-hydroxyjasionone have been achieved, because their cyclooctanoid core represents an important target for the development of synthetic strategies to prepare eight-membered ring-containing compounds. This review focuses on these natural sesquiterpenoids and their biological activities and synthesis, and aims to provide a foundation for further research of these interesting compounds.

## 1. Introduction

In the past thirty years, nature has played a significant role in the discovery of new drugs. Meanwhile, many natural products with interesting skeletons have been reported from terrestrial and marine sources [1,2,3,4]. Sesquiterpenoids, a group of naturally occurring 15-carbon isoprenoids, are widely distributed secondary metabolites in nature and show a vast array of interesting molecular architectures [5,6,7,8]. Among them, the unusual bicyclo [6.3.0] undecane sesquiterpenoids are one of the most representative. To the best of our knowledge, since dactylol was isolated from the sea hare *Aplysia dactylomela* in 1977 [9], only seven types of scaffolds of bicyclo [6.3.0] undecane sesquiterpenoids have been reported (Figure 1). Structurally, precapnellane (Figure 1a), asteriscane (Figure 1b), dumortane (Figure 1c), toxicodenane (Figure 1d), and capillosane (Figure 1e) featured four methyl groups on the 5-8 ring moiety, while jasionane (Figure 1f) and sinulane (Figure 1g) decorate the core with an isopropyl group. As the cyclooctanoid core represents an important and challenging target for the development of the methodology, necessary to prepare the eight-membered ring-containing compounds, some total syntheses of precapnellane-sesquiterpenoid [10,11,12,13,14,15,16,17,18,19,20,21,22,23,24,25,26,27,28,29], asteriscane-sesquiterpenoid [30,31,32,33,34,35,36], and jasionane-sesquiterpenoid [37] have been achieved. However, so far, no comprehensive review of these molecules has been published. To provide a foundation for further research, this review summarizes the structures, biological activities, and chemical synthesis of bicyclo [6.3.0] undecane sesquiterpenoids.

## 2. Four Methyl Type Bicyclo [6.3.0] Undecane Sesquiterpenoids

### 2.1. Precapnellane-Sesquiterpenoid

In 1977, the first bicyclo [6.3.0] undecane sesquiterpenoid, named dactylol (**1**, Figure 2), was reported in the Caribbean sea hare *Aplysia dactylomela* by Schmitz’s group [9]. Its absolute configuration was derived from the CD data of a substituted cyclopentanone degradation product, which is a chemical conversion compound. In its structure, as the cyclooctanoid core represents an important and challenging target [10] for preparing the eight-membered ring-containing compounds, dactylol, an exemplary member of this class, has been widely synthesized (in total) by synthetic chemists. In order to construct the necessary carbon–carbon bonds, both Paquette’s group and Gadwood’s group used a [3,3] sigmatropic rearrangement strategy from 1985 to 1987 [10,11,12]. On the other hand, in the Paquette route, a cycooctadiene ring was built from cycloheptane precursors by Friedel–Crafts cyclization, while the Gadwood route featured an anionic oxy–Cope rearrangement. Later, to solve the undesired isomer problem, originatingthe stereoselective functional group transformations, Feldman et al. developed concise and stereoselective novel [6π + 2π] intramolecular photocycloaddition to synthesize dactylol in 1989 and 1990 [13,14]. Then, in 2000, intramolecular 4 + 3 cycloaddition was achieved in the synthesis of dactylol by Harmata’s group [15,16]. In 1996, ring-closing metathesis was used to synthesize dactylol by Fürstner et al. [17], and Vanderwal’s group further developed this strategy [18]. Molandar’s team reported the concise, nonracemic synthesis of dactyol utilizing a novel [3 + 5] annulation approach in 1995 [19].

In 1978, dactylol (**1**), together with poitediol (**2**), whose structure was determined by X-ray crystallography, was found in the seaweed *Laurencia poitei* (Lamouroux) Howe by Fenical’s group [38]. Gadwood’s group and Vanderwal’s group also used their strategy to achieve the total synthesis of compound **2** [10,18,20,21].

One year later, Djerassi’s group reported the isolation of precapnelladiene (**3**) from the nonpolar fractions of the soft coral *Capnella imbricata* [39]. To establish its stereochemistry, Mehta and Murty synthesized precapnelladiene (**3**) through tricyclo [6.3.0.0^2,6^] undecane via ruthenium-catalyze oxidation [22,23]. The Claisen rearrangement strategy was also used as a prelude for the preparation of compound **3** by Paquette’s group and Petasis’s group [24,25,26]. Moore et al. also reported the total synthesis of precapnelladiene (**3**) through the oxy–Cope rearrangement [27,28]. In 2007, Iguchi’s group reported an enantioselective copper-catalyzed [2 + 2]-cycloaddition reaction to obtain compound **3** [29].

### 2.2. Asteriscane-Sesquiterpenoid

In 1985, San Feliciano and colleagues reported the first asteriscane type sesquiterpenoid, named asteriscanolide (**4**), from the hexane extract of plant *Asteriscus aquaticus* [40]. Its structure was determined by X-ray diffraction. The enantioselective total synthesis of asteriscanolide (**4**) was achieved in 1988 by Wender et al. based on nickel-catalysed intramolecular [4 + 4] cycloaddition [30]. In 2000, the Michael–Michael reaction sequence and ring-closing metathesis were used as the key bond-forming operations to synthesize compound **4** by Paquette’s group [31], and in the same year Krafft and colleagues published on the total synthesis of (±)-asteriscanolide by a intermolecular Pauson–Khand [2 + 2 + 1] cycloaddition reaction as the key transformation [32,33]. Additionally in 2000, Snapper’s group developed an efficient synthesis of (±)-**4** using sequential intramolecular cyclobutadiene cycloaddition, ring-opening metathesis, and Cope rearrangement reactions [34]. In 2011, Yu’s group described the total synthesis of (+)-**4** using a chiral ene-vinylcyclopropane substrate induced Rhodium(Ι)-catalyzed [(5 + 2) + 1] cycloaddition reaction to construct a 5-8 ring core [35,36].

Additionally in 1985, two asteriscane analogues 1,10,7,8-tetradehydro-asteriscanolide (**5**, Figure 3) and methyl-3α,5α,8α,10α*H*-asteriscan-15-oate (**6**, Figure 3) were found in the species *A. graveolens* by Bohlmann’s group [41].

In 1995 and 1999, in the plant *Lippia integrifolia*, a traditional medicine of north and central Argentina, Catalán’s group and König’s group reported two new asteriscane sesquiterpenes 3α-hydroxy-6-asteriscene (**7**, Figure 3) [42] and asterisca-3(15),6-diene (**8**, Figure 3) [43], respectively.

The first marine originating asteriscane sesquiterpene, asterisca-2(9),6-diene (**9**, Figure 3), was isolated from the marine animal aeolid nudibranch *Phyllodesmium magnum* by Guo et al. in 2011 [44]. Based on the predator–prey relationship between *P. magnum* and the soft coral genus of *Sinularia*, the source of compound **9** may be soft coral [44,45,46].

In 2013, fourteen new asteriscane-type sesquiterpenoids (capillosananes A-N) (**10**–**23**, Figure 3) (the first large-scale discovery of bicyclo [6.3.0] undecane sesquiterpenoids) were found in the soft coral *S. capillosa* by Lin’s group [45]. Their absolute configurations were determined by Mosher’s method (compounds **10**, **11**, **14**, **19**), CD rules (compounds **13**, **16**–**18**, **23**), ECD calculation (compounds **16** and **17**), biogenetic consideration (compound **15**), and chemical conversion (compounds **10**, **16**, **20**, **22**). In bioassay tests, these compounds were inactive against HCT-8, HePG2, BGC-823, A549, and SKOV3 human humor cell lines. Nevertheless, compounds **11** and **18** exhibited weak in vitro inhibitory effects on inflammation-related TNF-α. Compound **10** showed potent antifouling activity against *Balanus amphitrite* with an IC_50_ value of 9.70 μM, whereas the value of compound **18** was 54 μM. A capillosanane D derivative (also from the genus of *Sinularia* (*S. verruca*)), named deoxocapillosanane D (**24**, Figure 3), was isolated by Yan and coworkers in 2016 [46], but it showed no anti-HIV-1 or anti-inflammatory activities.

In the search for bioactive compounds in the roots of *Cynanchum wilfordii*, two new asteriscane type sesquiterpenoid wilfolides A and B (**25** and **26**, Figure 3) were isolated and identified by Zhao’s group in 2015 [47]. Their absolute configurations were elucidated by X-ray crystallography with Cu Kα radiation. Compound **25** exhibited weak inhibitory effect against acetylcholinesterase.

A chemical examination of the aerial parts of the *Asteriscus graveolens* subsp. *stenophyllus*. by León et al. in 2016 resulted in the isolation of asteriscanolidenol (**27**, Figure 3), a new sesquiterpene lactone of the asteriscanolide type [48]. Unfortunately, this lactone showed no cytotoxicity effects against the HL-60 and MOLT-3 leukemia cell lines.

### 2.3. Dumortane-Sesquiterpenoid

The first dumortane sesquiterpenoid, dumortenol (**28**, Figure 4), was isolated from the diethyl ether extract of Argentinian liverwort *Dumortiera hirsuta* by Toyota and coworkers in 1997 [49]. The stereochemical assignments of **28** were clarified by X-ray crystallographic analysis (crystals were obtained from the methanol solution). In 1999, their reinvestigation of a new collection of Argentine *D. hirsuta* led to the isolation of two new dumortane derivatives (**29** and **30****)**, together with nor-dumortane sesquiterpene **31** [50].

Compound **28** was also obtained from soft coral *Sinularia capillosa* by Duh’s group in 2010, which is the first report of this type of sesquiterpenoid in a marine organism [51]. In 2014, chemical examination of the same soft coral species was carried out by Lin’s group and resulted in the isolation of two dumortane analogues, capillosananes W and X (**32** and **33**, Figure 4) [52]. The absolute configuration of **32** was determined via the CD data of the in situ complex of the tertiary alcohol with Rh_2_(OCOCF_3_)_4_ by applying the bulkiness rule; **33** was assumed to be the same as **32** from biogenetic consideration. Bioactive assays indicated that compound **32** has anti-inflammatory effects with an inhibitory rate of 34% (at 10 μM, the positive control NK007 with inhibitory rates of 46% at 100 nM).

In 2014, Shen’s group reported only one microorganism-originated bicyclo [6.3.0] undecane sesquiterpenoid, named tuberculariol D from a mutant strain G-444 of *Tubercularia* sp. TF5 isolated from the inner bark of *Taxus mairei* (**34**, Figure 4) [53]. At 30 μg/disc, this compound exhibited no antifungal activities against *Candida albicans*.

### 2.4. Toxicodenane-Sesquiterpenoid

In 2013, toxicodenanes C (**35**, Figure 5) was obtained from the dried resin of *Toxicodendron vernicifluum* by Cheng’s group [54]. This compound showed significantly inhibitory effects with a dose- and time-dependent relationship on fibronectin, collagen IV, and IL-6 in high-glucose-induced mesangial cells, which means that it has potential in treating diabetic nephropathy.

### 2.5. Capillosane-Sesquiterpenoid

One year later, Lin’s group reported capillosananes S and T (**36** and **37**, Figure 5) from the soft coral *S. capillosa* [52]. The absolute configuration of compound **36** was determined by the octant rule for cyclopentenones and further supported by the ECD method; compound **37** was only determined by the octant rule. Unfortunately, these compounds were not active in cytotoxic (HCT-8, HePG2, BGC-823, A549, SKOV3) and pathogenetic microorganism assays.

## 3. Isopropyl Type Bicyclo [6.3.0] Undecane Sesquiterpenoids

Isopropyl type bicyclo [6.3.0] undecane sesquiterpenoids are rare in nature. By the end of August 2019, only 5 isopropyl type bicyclo [6.3.0] undecane sesquiterpenoids (far fewer than the four methyl type) have been reported. This may be due to the difficulty for diverse biogenic pathways to form a 5-8 fused ring in nature. As shown in Scheme 1, the plausible biosynthetic pathways of bicyclo [6.3.0] undecane sesquiterpenoids were proposed. The carbon skeletons of the four methyl type bicyclo [6.3.0] sesquiterpenoids can be carried out by cyclization reaction and Wagner–Meerwein rearrangements from the farnesyl diphosphate (FPP) [54,55]. However, an unexpected cycloheptane to cyclooctane ring expansion process [56], a key step, may determine the occurrence rate of isopropyl type bicyclo [6.3.0] undecane sesquiterpenoids.

### 3.1. Jasionane-Sesquiterpenoid

The first isopropyl-branched bicyclo [6.3.0] undecane sesquiterpenoid, named 11-hydroxyjasionone (**38**), was isolated from the aerial parts of the *Jasonia montana* plant by Ahmed et al. in 1988 [57]. It was first synthesised in 1994 by Trost and Parquette, using a TMM (2-(1-(trimethylsilyl)-1-cyclopropyl) allyl pivalate) cycloaddition strategy through fragmentation of the [3.3.0] system to form a 5-8-fused bicyclic core [37].

In the following year, Rustaiyan and coworkers isolated a new jasionane type sesquiterpene lactone tehranolide (**39**, Figure 6) from the aerial parts of *Artemisia diffusa* [58]. As there is an endoperoxide pharmacophore like the antimalarial agent artemisinin in the molecule, some bioactivity test works on compound **39** [59,60,61,62,63,64,65] and the fractions containing it [66,67] have been done by Iranian scientists. In summary, tehranolide (**39**) has a variety of biological activities, including modulating the immune response by reducing regulatory T cell [59,60], inhibiting proliferation of MCF-7, HeLa, and K562 cells [61,62,63], against chemical mutagens in *Salmonella* strains [63], and inhibiting the growth of *Plasmodium falciparum* [64,65].

In 2013, chemical investigations of the aerial parts of another *Artemisia* specie *A. vestita* by Tian et al. led to the isolation of two new jasionane sesquiterpenes: Arvestolides B and C (**40** and **41**, Figure 6) [68]. The absolute configuration of compound **40** was determined by single-crystal X-ray diffraction with Cu Kα radiation, and the absolute configuration of compound **41** was the same as compound **40** based on biogenetic consideration and comparing its optical rotation with **40**. Compound **40** showed a moderate inhibitory effect on lipopolysaccharide (LPS)-induced nitric oxide (NO) production in BV-2 microglial cells.

### 3.2. Sinulane-Sesquiterpenoid

In 2018, Li’s team found the first example of marine-originated isopropyl type bicyclo [6.3.0.] undecane sesquiterpenoid sinuketal (**42**, Figure 6) from soft coral *Sinularia* sp. [56]. Its relative and absolute configurations were determined on the basis of the NOESY spectrum in combination with a conformational analysis, density functional theory-NMR, and the TDDFT/ECD method. Compared with the similar analogue compound **39**, the positions of the methyl and isopropyl groups of compound **42** were quite different. The biological activity tests showed that **42** displayed antiviral activities (against influenza A viruses H1N1 and PR8, with IC_50_ values of 172 and 443 μM, respectively), weak cytotoxic activities (toward Jurkat, MDA-MB-231, and U2OS cell lines), mild in vitro antimalarial activity (against *Plasmodium falciparum* 3D7), as well as mild inhibitory acetylcholinesterase activity.

## 4. Conclusions

Bicyclo [6.3.0] undecane sesquiterpenoids are relatively rare in nature. From 1977 to 2018, only approximately 42 compounds with this unique scaffold were reported in terrestrial plants of the genera *Asteriscus*, *Lippia*, *Cynanchum*, *Dumortiera*, *Toxicodendron*, *Jasonia,* and *Artemisia*, marine organisms of the genera *Aplysia*, *Laurencia*, *Capnella*, *Sinularia,* and *Phyllodesmium*, and a mutant bacteria of *Tubercularia* sp. Their absolute configurations were determined by X-ray diffraction, Mosher’s method, CD rules, ECD calculaion, biogenetic consideration, and chemical conversion methods. They can be classified into seven different types. Structurally, precapnellane, asteriscane, dumortane, toxicodenane, and capillosane feature four methyl groups on the 5-8 ring moiety, while jasionane and sinulane decorate the core with an isopropyl group. How the isopropyl type bicyclo [6.3.0] undecane sesquiterpenoids were formed remains a mystery, though this question may be resolved by a biosynthesis study. Because of the broad bioactivities and synthetic challenges of the cyclooctanoid core, some total synthesis works on dactylol (**1**), poitediol (**2**), precapnelladiene (**3**), asteriscanolide (**4**), and 11-hydroxyjasionone (**38**) were achieved to develop the methodology to prepare the cyclooctane-containing compounds. Unique structural molecules continue to be a rich source for lead compound discovery.

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
