# Peer review of "Bicyclo [6.3.0] Undecane Sesquiterpenoids: Structures, Biological Activities, and Syntheses"

_molecules, 2019, doi:10.3390/molecules24213912_

Round 1
Reviewer 1 Report
The manuscript brings some interesting view on this type of sesquiterpenoids. It needs to present data in more detail, try to go a little bit deeper in the discussion.
Indeed, some minor should issues be observed.
-first person should be avoid all over the text, from the abstract.
-Keywords should be revised, these next could be changed by other that bring more interest: structures; biological activities; syntheses.
-structures should be numbered in sequence as they appear on the text. Maybe the insertion of a new first figure with the representative sesquiterpenoids could be a good choice.
-Figure 1 is poorly reported on text. It should be better explained or removed.
-Line 71, harmata; Line 81, respectively; Line 83, nonpoplar; Line 84, first total synthesized...the manuscript should be carefully revised. Many other parts show incomprehensible text.
-Text items, such as 2.2 and 3, should not initiate with a figure.
-Stereochemistry should be revised all over the text. Methyl and hydroxyl groups at Figure 3 structures are some examples.
-How scheme 1 was obtained?
-The text could be improved with better description of the papers cited. As an example, the many references (59-67) poorly described at lines 192-193 compared with reference 15, described all over the last paragraph and with biological activities detailed at lines 209-213.
Author Response
Point 1: first person should be avoid all over the text, from the abstract.
Response 1: line 24 and line 46, the first person statement of “our reported” were deleted; lines 52-54, the sentence “In the present review, we will summarize the developments of their structures, biological activities and synthesis, and aims to provide a foundation for further research” was rewritten as “To provide a foundation for further research, this review summarizes the structures, biological activities and chemical synthesis of bicyclo [6.3.0] undecane sesquiterpenoids”; line 207, the first person statement “our team” were changed to “Li’s team”
Point 2: Keywords should be revised, these next could be changed by other that bring more interest: structures; biological activities; syntheses.
Response 2: As this manuscript is a review, these keywords summarize the whole text, and are easily searched and browsed. These words may be reasonable to be reserved.
Point 3: Structures should be numbered in sequence as they appear on the text. Maybe the insertion of a new first figure with the representative sesquiterpenoids could be a good choice.
Response 3: When writing this manuscript, we have considered this problem, but it’s difficult to solve. Inspired by Point 4, in order to explain Figure 1, replacing structures (in the introduction part) to scaffolds can solve the problem (lines 44-51). The figure with the representative sesquiterpenoids is in the Graphical Abstract part.
Point 4: Figure 1 is poorly reported on text. It should be better explained or removed.
Response 4: line 42-47, Figure 1 was explained.
Point 5: Line 71, harmata; Line 81, respectively; Line 83, nonpoplar; Line 84, first total synthesized…the manuscript should be carefully revised. Many other parts show incomprehensible text.
Response 5: line 73, the incorrect writing “harmata” were corrected as“Harmata”; line 81, the incomprehensible text “respectively” were corrected as “Gadwood’s group and Vanderwal’s group also used their strategy to achieve the total synthesis of compound 2, respectively”; Line 83, the incorrect writing “nonpoplar” were corrected as “nonpolar”; Line 85, the statement of “first total synthesized” were corrected as “synthesized”. The manuscript was carefully revised and checked by a professional English editing service (Picture 1).
Point 6: Text items, such as 2.2 and 3, should not initiate with a figure.
Response 6: line 104 and line 181, the positions of Figure 3 and Scheme 1 were moved.
Point 7: Stereochemistry should be revised all over the text. Methyl and hydroxyl groups at Figure 3 structures are some examples.
Response 7: The stereocenters, which have two stereodescriptors, were corrected as only one stereodescriptor all over the text.
Point 8: How scheme 1 was obtained ?
Response 8: The plausible biosynthetic pathway of the four methyl type bicyclo [6.3.0] undecane sesquiterpenoids was proposed according to plausible biosynthetic pathway of toxicodenanes C (doi:10.1021/ol4014415), combinated with construction mechanisms (Pages:14-16) and biosynthetic pathway of sesquiterpenes (Pages: 210-217) in book “Medicinal Natural Products: A Biosynthetic Approach”. The plausible biosynthetic pathway of the isopropyl type bicyclo [6.3.0] undecane sesquiterpenoids was obtained according to the biogenetic pathway of sinuketal proposed by our team. “As shown in Scheme 1, a biogenetic pathway of 1 (sinuketal) can plausibly be retrospect to the isodaucyl cation, the biosynthetic precursor of isolated (7), which can be traced biogenically back to thegermacryl cation and farnesyl diphosphate (FPP). The carbocation of the isodaucyl cation can be discharged by enzyme-catalyzed cyclization to generate intermediate A, rather than by quenching with a nucleophile to form compound 7. Then, oxidation of intermediate A yields intermediate B, which subsequently undergoes oxygen-mediated hydroperoxidation and concerted ring expansion to form hydroperoxide intermediate C. Finally, the conversion from intermediate C into 1 could be carried out by intramolecular semi-acetalization reaction and Wagner–Meerwein 1,3-hydride shift” (doi:10.3390/md16040127).
Point 9: The text could be improved with better description of the papers cited. As an example, the many references (59-67) poorly described at lines 192-193 compared with reference 15, described all over the last paragraph and with biological activities detailed at lines 209-213.
Response 9: Lines 192-196, references (59-67) were better described.

Reviewer 2 Report
This manuscript is a review of the naturally occurring sesquiterpenes that possess the bicyclo[6.3.0]undecane ring system. The manuscript briefly describes the isolation, structure determination, bioactivities and synthetic efforts on this group of compounds. Overall, the review provides a useful history of this area, however, there are some errors in the terminology used, and a significant revision is necessary to improve the writing before publication. A partially corrected draft is included with this review, but addition revision will be required beyond these suggested changes.

Author Response
Point: This manuscript is a review of the naturally occurring sesquiterpenes that possess the bicyclo[6.3.0] undecane ring system. The manuscript briefly describes the isolation, structure determination, bioactivities and synthetic efforts on this group of compounds. Overall, the review provides a useful history of this area, however, there are some errors in the terminology used, and a significant revision is necessary to improve the writing before publication. A partially corrected draft is included with this review, but addition revision will be required beyond these suggested changes.
Response: Thank you very much for reading our manuscript so carefully. The manuscript was carefully revised and checked by a professional English editing service (Picture 1).

Round 2
Reviewer 1 Report
Suggestions were observed and most of the modifications were performed, even not in a deeper way.
Reviewer 2 Report
The revised submission addresses many of the concerns raised in the previous draft.
There are only a few minor issues for the authors to consider. Here are the suggested corrections to the revised text:
Page 1, line 19: remove the word “showed”
Page 2, line 69: “originating the…” should probably be “originating from the…”
Page 6, line 184: “Trostand Parquette” should be “Trost and Parquette”
Page 7, line 216: “relativelyrare” should be “relatively rare”